# A cohort analysis of survival and outcomes in severely anaemic children with moderate to severe acute malnutrition in Malawi

Thandile Nkosi-Gondwe[1,2]*, Job Calis[2,3,4], Michael Boele van Hensbroek[2,3,4,5], Imelda Bates[5], Björn Blomberg[6,7], Kamija S. Phiri[2]

**1** Centre for International Health, Department of Global Public Health and Primary Care, University of Bergen, Bergen, Norway, **2** School of Public Health & Family Medicine, College of Medicine, University of Malawi, Blantyre, Malawi, **3** Liverpool–Wellcome Trust Clinical Research Programme, College of Medicine, Blantyre, Malawi, **4** Emma Children's Hospital, The Global Child Health Group, Academic Medical Center, University of Amsterdam, Amsterdam, The Netherlands, **5** Liverpool School of Tropical Medicine, Liverpool, United Kingdom, **6** Department of Clinical Science, University of Bergen, Bergen, Norway, **7** Norwegian National Advisory Unit on Tropical Infectious Diseases, Haukeland University Hospital, Bergen, Norway

* thandile_nkosi@yahoo.com

## Abstract

### Introduction

Moderate to severe acute malnutrition (SAM/MAM) and severe anaemia are important and associated co-morbidities in children aged less than five years. Independently, these two morbidities are responsible for high risk of in-hospital and post-discharge deaths and hospital readmissions. The primary objective of this study is to investigate the risk of death among severely anaemic children with moderate to severe acute malnutrition compared to children with severe anaemia alone.

### Methods

This was a retrospective analysis of data collected from a large prospective study that was investigating severe anaemia in children aged less than 5 years old. The study was conducted at Queen Elizabeth Central Hospital in Blantyre and Chikhwawa district hospital in southern Malawi. Children aged less than five years old; with severe anaemia were screened and enrolled. Each child was followed up for eighteen months at one, three, six, twelve and eighteen months after enrolment. Data were analysed using STATA 15.

### Results

Between July 2002 and July 2004, 382 severely anaemic children were enrolled in the main study. A total of 52 children were excluded due to missing anthropometric data. Out of the 330 included, 53 children were moderately to severely malnourished and 277 were not. At the end of the 18-month follow period, 28.3% of children with MAM/SAM died compared to 13% of children without MAM/SAM (RR 2.1, CI 0.9–4.2, p = 0.03). Similarly, children with moderate to severe malnutrition reported a significantly higher number of malaria infection cases (33.9%) compared to children with severe anaemia alone (27.9%, p = 0.02).

**Data Availability Statement:** All data can be accessed at:https://figshare.com/articles/dataset/

Moderate_to_severe_malnutrition_in_severe_
anemia_study_/13061447.

**Funding:** The authors received no specific funding
for this work.

**Competing interests:** The authors have declared
that no competing interests exist.

However, the number of hospitalizations and recurrence of severe anaemia was similar and not statistically significant between the two groups (RR 0.8 (0.4–1.4), p = 0.6 and RR 1.1 (0.3–2.8), p = 0.8).

## Conclusion

Among children with severe anaemia, those who also had moderate to severe malnutrition had a twofold higher risk of dying compared to those who did not. It is therefore crucial to investigate acute malnutrition among severely anaemic children, as this might be treatable factor associated with high mortality.

## Introduction

Malnutrition is a complex and multifactorial condition that results from deficiencies, excesses, or imbalances in a person's intake of energy and/or nutrients and presents in two broad forms; undernutrition which includes wasting, stunting and underweight; and micronutrient deficiencies and obesity [1]. Globally, it is estimated that 52 million children under five are wasted, 75% of which are in low- and middle-income countries (LMIC) [2]. The 2016 Malawi demographic and health survey (DHS) reported that 37% of children under five years of age are stunted while 12% are underweight [3], and mortality of up to 42% has been reported among children hospitalised with severe acute malnutrition (SAM) [4, 5]. Such high mortality rate has been attributed to comorbidities such as infections, including HIV, as well as micronutrient deficiencies including anaemia that often affect children with SAM [6, 7].

Severe anaemia is one of the most common causes of admissions and mortality in Sub-Saharan Africa (SSA) and annually affects 9.6 million children globally [8, 9]. In Malawi, it was found that children with severe anaemia alone have a tenfold risk of dying within 18 months after the initial episode compared to children from the hospital and community who did not have severe anaemia [10]. Similar findings have been reported in Kenya and Uganda [11, 12]. Causes of severe anaemia are multifactorial and vary in different settings. In addition to infectious diseases, genetic factors and malignancies, nutritional deficiencies are a major factor [13–16].

Severe anaemia and any form of malnutrition are common and associated co-morbidities. Anaemia is the most common manifestation of micronutrient deficiency in malnourished children under 5 years old [17], with up to 67% of severely malnourished children found to be severely anaemic [18, 19]. In Malawi, up to 63% of malnourished children have some form of anaemia signified by a Haemoglobin (Hb) level of <11.0g/dl and of these, 22% are moderate to severely anaemic (Hb <7g/dl). On the other hand, 15.8% of severely anaemia children are found with SAM [2].

Severe anaemia is an important co-morbidity and determinant in the recovery of children with malnutrition, so much so that WHO recommends that children with kwashiorkor or marasmus should be assumed to be severely anaemic [20]. Many studies have reported the in-hospital and post-discharge mortality outcomes among children with SAM [5, 21], and children with severe anaemia [11, 22] separately. However, the added risk of dying when a child has both of these conditions; which we predict would be high has not previously reported. As MAM/SAM is a potentially treatable risk factor [3]; we aimed to evaluate its impact on mortality in severely anaemic children.

## Materials and methods

This study was approved by the ethics committees of the College of Medicine, University of Malawi and the Liverpool School of Tropical Medicine, United Kingdom. In the SEVANA cohort study, children with severe anaemia and aged less than five years old were enrolled from Queen Elizabeth Central hospital and Chikhwawa district hospital in southern Malawi. Enrolment procedures have been extensively described elsewhere [14]. In summary, 382 children admitted with severe anaemia (defined as a haemoglobin concentration <5.0 g/dL) were enrolled and matched with one community and one hospital control. Each child was then followed up for 18 months.

In the present study, verbal and written informed consent was obtained from the legal guardians and children with severe anaemia who had a recorded weight, length and met the WHO classification of moderate to severe malnutrition defined as a child whose weight-for-length is less than <2 of the Z-scores and weight for age less that -2 of the z-scores [23]. An additional inclusion criterion was documented haemoglobin at enrolment and a documented date of outcome and status (died, completed, lost-to-follow up or withdrawn).

Information about the child's age, sex, residence, number of living and dead siblings, 24-hour dietary recall, family history of sickle cell disease (SCD), jaundice, bloody stool and urine, blood transfusion in the last two months, being on any medication, HIV infection and other previous medical history were obtained from the legal guardians at enrolment. Additional information included guardian's age, occupation and education level.

Other data points collected include physical examination findings at enrolment and laboratory records, which included parasitology, microbiology, haematology and biochemistry.

### Power calculation

We did a power calculation to evaluate the statistical power of our study due to the limited sample size. Using open EPI version 3 (www.openepi.com), we computed the 95% two-sided confidence interval, risk ratio (2.1) and the number of children sampled in each study group (53 with MAM/SAM versus 275 without SAM/MAM). We found that our analysis gave us a power of 76.4%.

### Statistical analyses

Data were coded, entered and analysed using STATA 15 (StataCorp, College Station, Texas, USA). Categorical variables have been summarized as frequencies and proportions, while continuous variables as means with standard deviations and medians with the interquartile ranges (IQR) reported. Death was our primary outcome. We examined risk of dying by calculating mortality rates in the children with moderate to severe acute malnutrition and those without. We measured time to death by survival analysis, using Kaplan-Meier curves to compare the probability of death between the two groups over the 18-month study period. Significance was calculated with a log-rank test. Incidence rates for the composite outcomes i.e. re-hospitalization, malaria and severe anaemia recurrence were also calculated for each group. For the malaria incidence rate, the time at risk was calculated by subtracting 14 days from the child-years follow-up with each case of clinical malaria treated with Lumefantrine-Artemether (AL). P values and 95% confidence intervals have also been included.

## Results

Of the 1141 under-5 children enrolled in the SEVANA study between July 2002 and July 2004, 382 were severe anaemic cases admitted to the paediatric wards of QECH and Chikhwawa

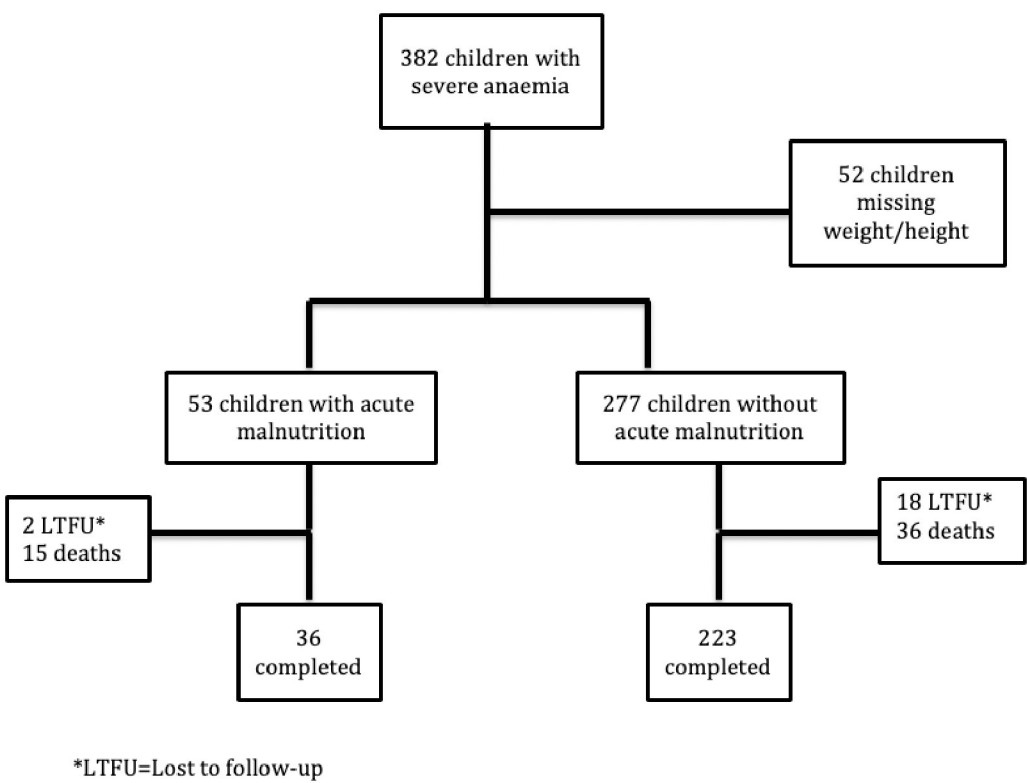

**Fig 1. Study flowchart.**

district hospital and 759 were hospital or community controls without severe anaemia. Of the 382 children, 330 had their weight and heights measured and were included in the final analysis. A total of 53 children had a weight for height z-scores ≤ -2 (moderate to severe acute malnutrition) and 277 had a weight for height z-scores > -2 (not malnourished). During the follow up period, twenty children were lost to follow up, 51 died and 259 children completed the study (Fig 1).

The baseline characteristics and examination findings of study participants was comparable between the two groups (Table 1). Moderate to severely malnourished children were significantly older with mean age, 24.3 months (SD 12.1) compared to 19.5 months (SD 12.4). The mean number of days hospitalised was not significantly different, 4.9 days (SD 8.1) compared to 3.9 days (SD 4.1). A higher proportion of non-malnourished children resided in a rural location, 52.4% compared to 45.3% in the malnourished group. The majority of parents of the children had received some formal education and about half of them had employment. A total of 34.0% of malnourished children had a sibling who died compared to 28.2% among children without (p = 0.05).

A total of 196 children (59.4%) had a positive blood smear for *P. Falciparum* malaria infection on admission, and of these 32 (60.4%) were children with MAM/SAM and 164 (59.2%) were children without MAM/SAM. C-reactive protein (CRP), a common marker of inflammation was raised (≥10Mg/L) among 83% of the children with MAM/SAM compared to 82.7% of those without MAM/SAM. A total of 27.3% of all the children had iron deficiency. A higher proportion of children with vitamin A and vitamin B12 deficiency were those without MAM/SAM.

**Table 1. Baseline characteristics of study participants.**

| Characteristic | Moderate to severe malnutrition | Severe anaemia alone | P-value |
|---|---|---|---|
| | n (%) | n (%) | |
| **Age <24 months** | **30 (56.6)** | **200 (72.2)** | **0.02[a]** |
| **Mean age in months n (SD)** | **24.3 (12.1)** | **19.5 (12.4)** | **0.01[a]** |
| Mean days in hospital n (SD) | 4.9 (8.2) | 3.9 (4.1) | 0.18 |
| Male gender | 23 (43.4) | 129 (46.6) | 0.67 |
| Rural location | 24 (45.3) | 145 (52.4) | 0.35 |
| **One or more dead siblings** | **18 (34.0)** | **78 (28.2)** | **0.05[a]** |
| Educated father | 35 (66.0) | 172 (62.1) | 0.72 |
| Teenage mother | 8 (15.1) | 71 (25.6) | 0.15 |
| **Uneducated mother** | **5 (9.4)** | **30 (10.8)** | **0.03[a]** |
| **Jobless parent** | **26 (49.1)** | **164 (59.2)** | **0.03[a]** |
| One dead parent | 5 (9.4) | 18 (6.5) | 0.72 |
| Previous blood transfusion | 9 (17.0 | 38 (13.7) | 0.53 |
| Recent antimalarial use | 31 (58.5) | 175 (63.2) | 0.72 |
| History of bloody stool | 6 (11.3) | 19 (6.9) | 0.45 |
| History of bloody urine | 1 (1.9) | 5 (1.8) | 0.97 |
| **Jaundiced** | **6 (11.3)** | **10 (3.6)** | **0.02[a]** |
| Splenomegaly | 33 (62.3) | 178 (64.3) | 0.49 |
| Raised CRP (≥10Mg/L) | 44 (83.0) | 229 (82.7) | 0.96 |
| Median CRP n (IQR) | 111.6 (65.2–183.2) | 93.6 (37.8–150.5) | 0.27 |
| Mean Haemoglobin n (SD) | 3.6 (0.8) | 3.4 (1.0) | 0.13 |
| Low Vitamin B12 (<118pmol/L) | 13 (24.5) | 71 (25.6) | 0.97 |
| Iron deficiency | 14 (26.4) | 76 (27.4) | 0.99 |
| Malaria infection at enrolment | 32 (60.4) | 164 (59.2) | 0.87 |
| HIV infected | 7 (13.2) | 29 (10.5) | 0.30 |
| CMV infection | 3 (1.1) | 0 (0.0) | 0.52 |
| EBV infection | 11 (20.8) | 63 (22.7) | 0.66 |
| Bacteraemia | 4 (7.6) | 36 (13.0) | 0.48 |
| Sickle cell disease | 1 (1.9) | 3 (1.1) | 0.88 |

[a] P-value which are significant at alpha = 0.05

During the 18-month study period, the mean observation days was 383 in the severely anaemic children with MAM/SAM and 456 days in the severely anaemic alone group respectively (Fig 2 and Table 2).

The cumulative proportions of children who died during the entire 18 month study period was 51 (15.4%) with 27 (8.2%) dying within one month of admission. Of the 51 deaths, 15 (28.3%) occurred in children with MAM/SAM compared to 36 (13.0%) who did not. The overall incidence rate of death with the 95% CI was 3.0 (2.2,4.0) children per 1000 person days observed. The incidence rates for death were 5.4 (3.0,9.8) and 2.6 (1.9,3.7) among children with MAS/SAM and those without respectively. This shows that children with MAM/SAM had a twofold risk of dying compared to children who has severe anaemia without MAM/SAM (RR 2.1; CI 0.9–4.2,p = 0.03). Similarly, severely anaemic children who were underweight had almost a 3-fold risk of dying compared to those with severe anaemia alone (RR 2.8; CI 1.5–5.2, p = 0.0006).

The survival curves for the two groups showed a statistically non-significant difference in the two mortality rates (log rank = 2.9, p = 0.098) (Fig 2). However, there was a significant

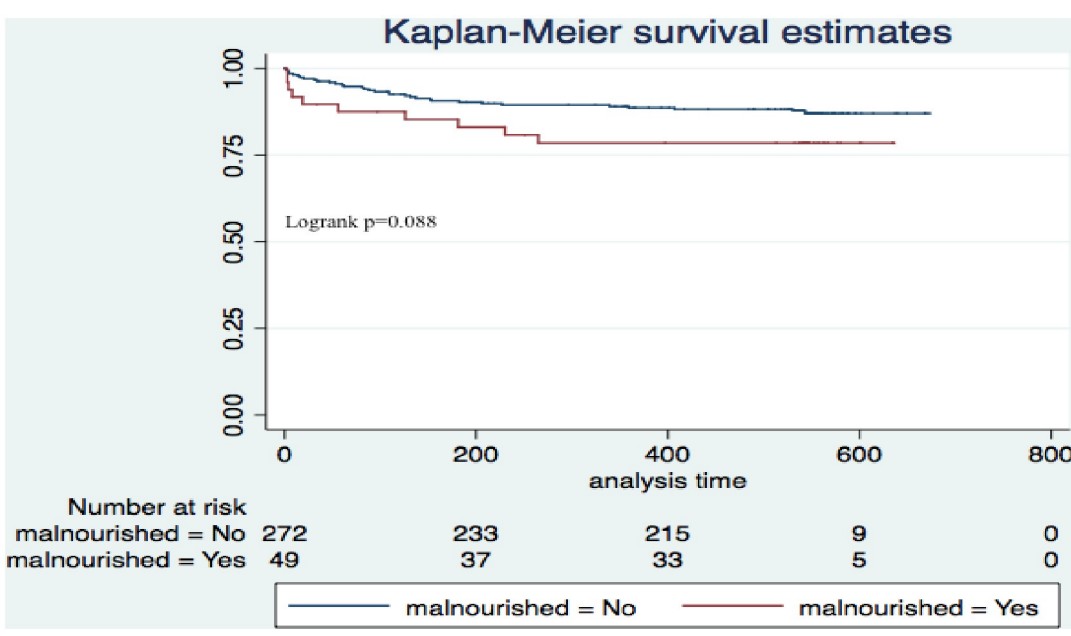

**Fig 2. Kaplan Meier survival curves of severely anemic children with moderate to severe acute malnutrition compared to those with severe anaemia alone.**

difference in mortality when we compared severely anaemic children who were underweight compared to those who were not, p = 0.001 (Fig 3).

During the follow up period, there were a total of 679 confirmed malaria cases, 16 of which were complicated malaria. There were significantly more children who reported malaria infection among those with MAM/SAM 106 (40.8%) compared to 573 (32.3%) (p = 0.01) with severe anaemia alone (IRR 1.3; CI 1.04–1.6, p = 0.02). There were more hospital readmissions among children who had severe anaemia alone compared to those with moderate to severe acute malnutrition (4.4% versus 3.5%), (IRR 0.8; CI 0.4–1.7,p = 0.62), but this was not statistically significant. In addition, the recurrence of severe anaemia was similarly low between children with moderate to severe malnutrition compared to those who had severe anaemia alone (IRR 1.1, CI 0.3–2.8, p = 0.8) over the entire study period.

**Table 2. Post-discharge morbidity and mortality among severely anaemic children with moderate to severe malnutrition compared to those with severe anaemia alone.**

| Event | Moderate to severe malnutrition | | Severe anaemia alone | | | |
|---|---|---|---|---|---|---|
| | Total events | Incidence rate (1000 person- days) | Total events | Incidence rate (1000 person- days) | Rate ratio | p-value |
| | N = 53 | | N = 277 | | n (CI) | |
| Deaths n (%) | 15 (28.3) | 5.4 | 36 (13.0) | 2.6 | 2.1 (0.9, 4.2) | 0.03[a] |
| Hospitalisation n (%) | 9 (3.5) | 1.5 | 76 (4.4) | 1.8 | 0.8 (0.4, 1.7) | 0.62 |
| Malaria incidence n (%) | 106 (40.8) | 17.3 | 573 (32.3) | 13.0 | 1.3 (1.04, 1.6) | 0.02[a] |
| Severe anaemia recurrence n (%) | 5 (1.6) | 0.8 | 32 (1.6) | 0.7 | 1.1 (0.3, 2.8) | 0.81 |

[a] P-value which are significant at alpha = 0.05

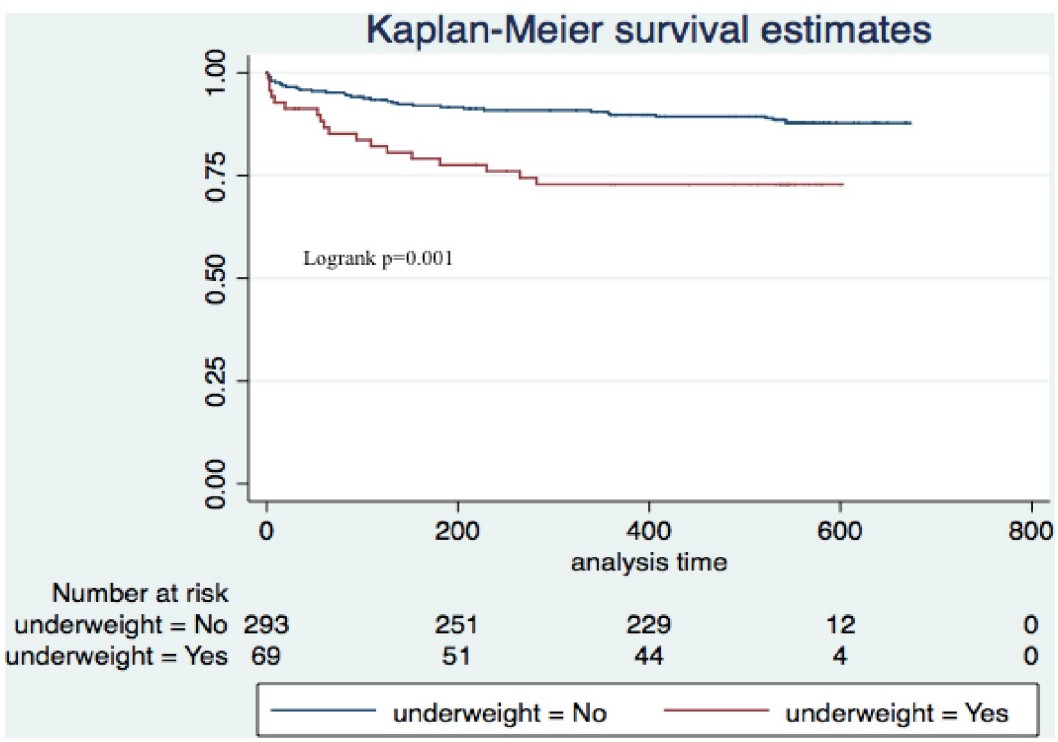

**Fig 3. Kaplan Meir survival estimates comparing severely anaemic children who are underweight and those that have severe anaemia alone.**

## Discussion

To our knowledge, no cohort studies investigating mortality outcomes in children with both severe anaemia and MAM/SAM have been conducted in SSA. We found that severely anaemic children with MAM/SAM are two times more likely to die compared to severely anaemic children without MAM/SAM during 18 months follow-up. High mortality rates have been reported among hospitalized children with severe anaemia and MAM/SAM separately in other African countries [24]. However, our findings show a much higher mortality rate than those reported in studies in children with severe anaemia in Ethiopia and Gambia [22, 25]. This higher mortality could be attributed to the fact that children included in our study had both conditions, which independently are major causes of mortality in children.

There are few cohort studies that have investigated mortality outcomes in children with both severe anaemia and MAM/SAM. Most of these have reported varying ranges of the burden of these two co-morbidities but there is limited data on impact. One study in Ethiopia reported that there was no significant difference in recovery among severe malnourished children with anaemia compared to those without anaemia [26]. On the other hand another study in Ethiopia found that children with SAM and anaemia had less chance of recovering compared to those who had SAM and no anaemia [27], which was similar to findings of a study in Kenya (Kwambai et al, Unpublished). However, this study did not have a comparison group. These findings are important because children admitted to hospital with severe anaemia are not routinely screened for MAM/SAM [28–30]. This has the implication that these children will not be checked for MAM/SAM, as it may be a treatable factor associated with high mortality.

The interplay between MAM/SAM, severe anaemia and the risk of mortality is multifactorial. It is believed that malnutrition lowers immunity, which leads to susceptibility to infections [31]. In addition, it is also possible that children with MAM/SAM are more likely to have micronutrient deficiency leading to cell damage that makes infections worse and leads to poor outcomes. Studies have attributed poor recovery outcomes in children with either MAM/SAM or severe anaemia to many factors including infections such as malaria [32–34]. We found that there was significantly higher malaria incidence among severely anaemic children with moderate to severe acute malnutrition compared to those with severe anaemia alone. The association between malnutrition and malaria has been inconclusive and conflicting [34]. In malaria endemic regions, malaria infection is associated with anaemia although it may not be the primary cause of it [35]. This finding is similar to other studies in Cameroon, Ghana and Gambia, where under-nutrition was associated with increased risk of malaria-associated mortality and multiple malaria infections [36–38].

Compared to other studies among children with severe anaemia or malnutrition, we did not find significant differences in hospital re admissions or recurrent severe anaemia. We had few fewer re-hospitalisations to detect meaningful differences. Although we did not make associations with other risk factors, our findings are important for exploring interventions that may reduce the additional burden that exists among severely anaemic children with malnutrition.

Our study had limitations. Our sample size was limited and we collected data from existing data that are now 16 to 18 years old. This might have an effect on the findings and their interpretation within the current context. Considering that mortality is a relatively rare outcome and the number of deaths was small, the confidence intervals for the mortality risks are relatively wide. However, we were able to detect significant and meaningful differences in mortality between the two groups of children.

## Conclusions and recommendations

Severe anaemic children who also have moderate to severe malnutrition have a higher risk of death than those with severe anaemia alone, even after discharge from hospital. It is therefore crucial to carefully screen for acute malnutrition in children admitted with severe anaemia, as this may be a treatable factor associated with high mortality. Prospective cohort studies may be utilised to evaluate effects of interventions that may be used to reduce mortality among severely anaemic children with moderate to severe acute malnutrition.

## Acknowledgments

We wish to acknowledge the immense contributions of the entire SEVANA team for the data and the children who participated in the main trial.

## Author Contributions

**Conceptualization:** Job Calis, Michael Boele van Hensbroek, Imelda Bates, Björn Blomberg, Kamija S. Phiri.

**Formal analysis:** Thandile Nkosi-Gondwe.

**Funding acquisition:** Michael Boele van Hensbroek, Imelda Bates.

**Investigation:** Job Calis, Michael Boele van Hensbroek, Imelda Bates, Björn Blomberg.

**Methodology:** Björn Blomberg, Kamija S. Phiri.

**Supervision:** Björn Blomberg, Kamija S. Phiri.

**Writing – original draft:** Thandile Nkosi-Gondwe.

**Writing – review & editing:** Job Calis, Michael Boele van Hensbroek, Imelda Bates, Björn Blomberg, Kamija S. Phiri.

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
