## [Decision Letter · Decision Letter 0]

15 Jul 2020

PONE-D-20-12989

A cohort analysis of survival and outcomes in severely anaemic children with moderate to severe acute malnutrition in Malawi

PLOS ONE

Dear Dr. Gondwe,

Thank you for submitting your manuscript to PLOS ONE. After careful consideration, we feel that it has merit but does not fully meet PLOS ONE’s publication criteria as it currently stands. Therefore, we invite you to submit a revised version of the manuscript that addresses the points raised during the review process.

Please be sure to prepare a point-by-point response to the reviewers' comments. In general, they found the manuscript to be of interest but clarifications are required. The biggest concern has to do with the applicability of the findings to today's context. Are the authors able to cite evidence that MAM/SAM might go unnoticed in a severely anemic child who presents to a health clinic?

We look forward to receiving your revised manuscript.

Kind regards,

Laura E. Murray-Kolb

Academic Editor

PLOS ONE

Journal Requirements:

2. Please provide additional details regarding participant consent. In the ethics statement in the Methods and online submission information, please ensure that you have specified (1) whether consent was informed and (2) what type you obtained (for instance, written or verbal). If your study included minors, state whether you obtained consent from parents or guardians. If the need for consent was waived by the ethics committee, please include this information.

Reviewers' comments:

Reviewer's Responses to Questions

**Comments to the Author**

1. Is the manuscript technically sound, and do the data support the conclusions?

Reviewer #1: Yes

Reviewer #2: Partly

2. Has the statistical analysis been performed appropriately and rigorously? 

Reviewer #1: Yes

Reviewer #2: Yes

3. Have the authors made all data underlying the findings in their manuscript fully available?

Reviewer #1: No

Reviewer #2: Yes

4. Is the manuscript presented in an intelligible fashion and written in standard English?

Reviewer #1: Yes

Reviewer #2: Yes

5. Review Comments to the Author

Reviewer #1: PLOS ONE article review; Manuscript Number: PONE-D-20-12989

Title: A cohort analysis of survival and outcomes in severely anaemic children with moderate to severe acute malnutrition in Malawi

Comments

There are several places where it is not clear whether you are talking about malnutrition in general or MAM/SAM. Such statements should be made more explicit to avoid confusing a reader should your paper be published.

Abstract

Line#41: correct this typo” theses”

Line#44: “Add company name and location for “STATA 15”

Line#48-50: These two sentences should be made clearer. So 15 out of 53 SAM children died vs. 36 out of 275 severely anemic children? 53+275=328, but you had weight and height data for 330 children. 28.3% is a cumulative value across 18 months. What does 53 represent? Is this a cumulative value or the number of SAM children with anemia throughout the study period?

Introduction

After reference 2, “correct the typo in “It”

After reference 3, “…nearly half of all children under five years are

Malnourished…" Given the broad definition of malnutrition in your first sentence, be specific with what malnutrition represents here. For instance, stunting, Stunting + anemia?

Just before reference 8, are community controls here non-anemic healthy children? Anemic children, that are managed at the community level?

Just before reference 16, “… severely malnourished children…”, are these SAM children?

Statistical analysis

First sentence, see comments above under abstract.

The penultimate sentence, what is "AL"? Write the full name on the first appearance.

Page 8: second paragraph, 196 children out of 330? If so, 60.4% (n=118???). Is the total number of malnourished children, not 53?

Discussion

First sentence, does “… our setting ..” here mean Malawi? Sub-Saharan Africa?

The second sentence, add "are" between "malnutrition" and "two."

After reference 19, correct they typo “…higher mortality rate than that…" "that" to "those": Also recheck the references. Reference 21 is the Gambia, not Nigeria.

Page 13: penultimate sentence, correct typo in “Children”.

Second paragraph, “Unpublished, T. Kwambai”.

Check this study from Ethiopia, Int J Pediatr. 2020; 2020: 8406597.

doi: 10.1155/2020/8406597

References

Recheck your all to ensure consistent formatting and style.

Figure 1: Flowchart

52+(53+275) =380. You have not accounted for 2 children.

Figure 2 and Figure 3: Kaplan-Meier plots

The axes should be properly labelled.

Reviewer #2: The authors use old data (2002-2004) from a small sample (n=330) of hospitalized children under five years old in Malawi. Following these children for 18 months, they compare the death rate in children with severe anaemia to the death rate in children with severe anaemia plus at least moderate acute malnutrition (WHZ < -2). Compared to children with severe anaemia alone, children with severe anaemia plus MAM/SAM were twice as likely to die. The authors conclude by recommending that children who are hospitalized with severe anaemia should also be screened for MAM/SAM.

My primary concern is that the findings are not relevant in today’s context. I may be wrong, but it is hard to believe that hospitalized children are not screened for MAM/SAM. I would guess that weight and height are among the first things to be measured when any child is hospitalized. If a child presents with both MAM/SAM and anaemia, they will receive food and iron supplements, two different treatments. A child who presents with MAM/SAM may also be screened for anaemia. It is hard to believe that MAM/SAM would go completely unnoticed by doctors in today’s context. The authors need to address this and justify the relevance of the study using recent evidence. Other minor comments are below.

Abstract

The following sentence is unclear: “Under-five children with severe anaemia were screened and enrolled and of theses children with moderate to severe acute malnutrition; defined as weight-for-height Z-score <-2 were included.” Enrolled and included are distinct? Do you mean included in the analysis? And of course there is a bad typo, “theses”.

Don’t use acronym SAM for moderate to severe malnutrition. SAM is severe (HAZ < -3) and does not include moderate. You could say 53 children were identified as having MAM or SAM.

Introduction

The manuscript is missing line numbers.

The first sentence of the introduction is incorrect. Malnutrition goes much beyond intake. Inadequate or excess intake among other factors can lead to malnutrition.

Change ‘commonest’ to most common.

Methods

Power calculation: can you clarify what the study groups were and what the sample size was for each group? Is it the 53 with MAM/SAM and 275 without MAM/SAM?

What is the acronym AL? Albendazole?

Results

Table 1 – groups need to be labeled better. Severe anaemia + MAM/SAM vs. Severe anaemia alone

Discussion

Do not abbreviate severe anaemia as SA. Avoid abbreviations whenever possible.

The data are 16-18 years old, not 16 years old.

6. PLOS authors have the option to publish the peer review history of their article (what does this mean?). If published, this will include your full peer review and any attached files.

Reviewer #1: No

Reviewer #2: No

---

## [Author Response · Author response to Decision Letter 0]

12 Nov 2020

Dear PLOS ONE,

 Thank you for reviewing our manuscript titled “A cohort analysis of survival and outcomes in severely anaemic children with moderate to severe acute malnutrition in Malawi” for consideration for publication in your esteemed journal. We have revised our manuscript with considerations to the comments and suggestions from the reviewers and ensured that it meets PLOS ONE’s style requirements. We would like to confirm that informed verbal and written consent were obtained from the guardians of all the children who participated in the study and a statement regarding ethical considerations has been added under the methods section. In addition, there are no ethical or legal restrictions on sharing of our data set. The anonymized data set can be found on: 

https://figshare.com/articles/dataset/Moderate_to_severe_malnutrition_in_severe_anemia_study_/13061447

Please kindly update our Data availability statement to reflect the availability of data.

Responses to reviewers

Below are point by point responses to each comment raised by the reviewers. Responses are highlighted in bold.

1. There are several places where it is not clear whether you are talking about malnutrition in general or MAM/SAM. Such statements should be made more explicit to avoid confusing a reader should your paper be published.

It is correct that we are referring to MAM/SAM and we have made corrections throughout the manuscript.

2. Abstract

Line#41: correct this typo” theses”

This has been corrected.

3. Line#44: “Add company name and location for “STATA 15”

We have added the company name and location for “STATA 15”.

4. Line#48-50: These two sentences should be made clearer. So 15 out of 53 SAM children died vs. 36 out of 275 severely anemic children? 53+275=328, but you had weight and height data for 330 children. 28.3% is a cumulative value across 18 months. What does 53 represent? Is this a cumulative value or the number of SAM children with anemia throughout the study period?

This has been revised. The correct number of children without MAM/SAM was 277. Revisions to the text and the study flow chart (Figure 1) have been made accordingly.

5. Introduction

After reference 2, “correct the typo in “It”

This has been corrected.

6. After reference 3, “…nearly half of all children under five years are

Malnourished…" Given the broad definition of malnutrition in your first sentence, be specific with what malnutrition represents here. For instance, stunting, Stunting + anemia?

We have revised this in line 81-84. We have defined malnutrition as follow: it is a complex and multifactorial condition that results from deficiencies, excesses, or imbalances in a person’s intake of energy and/or nutrients and presents in two broad forms; undernutrition which includes wasting, stunting and underweight; and micronutrient deficiencies and obesity.

7. Just before reference 8, are community controls here non-anemic healthy children? Anemic children, that are managed at the community level?

Thank you very much. Community controls were children matched for age and sex from the community and hospital who did not have severe anemia. A revision has been made in live 94 to make it clearer.

8. Just before reference 16, “… severely malnourished children…”, are these SAM children?

This is correct. We have revised this to make it clearer.

9. Statistical analysis

First sentence, see comments above under abstract.

We have revised this with a full name of the company and location in line 180-181 under statistical analyses.

10. The penultimate sentence, what is "AL"? Write the full name on the first appearance.

AL is the abbreviation for Artemether-Lumefantrine and it has been written in full in line 192-193.

11. Page 8: second paragraph, 196 children out of 330? If so, 60.4% (n=118???). Is the total number of malnourished children, not 53?

This was 196 out of 330, representing 59.4% with a positive blood smear for P. Falciparum malaria infection on admission, and out of these, 32 ( 60.4%) were children with MAM/SAM and 164 (59.2 %) were children without MAM/SAM. Revisions have been made in line 216-218 to make it clearer.

12. Discussion

First sentence, does “… our setting ..” here mean Malawi? Sub-Saharan Africa?

“Our setting” means Sub-Saharan Africa. We have revised this to reflect the same in line 287.

13. The second sentence, add "are" between "malnutrition" and "two."

This has been corrected in line 288.

14. After reference 19, correct they typo “…higher mortality rate than that…" "that" to "those": Also recheck the references. Reference 21 is the Gambia, not Nigeria.

This has been corrected in line 291. The references have also been corrected in line 292.

15. Page 13: penultimate sentence, correct typo in “Children”.

This has been corrected.

16. Second paragraph, “Unpublished, T. Kwambai”.

Check this study from Ethiopia, Int J Pediatr. 2020; 2020: 8406597.

doi: 10.1155/2020/8406597

Thank you very much. We have reviewed this reference and included it in our text.

17. References.

Recheck your all to ensure consistent formatting and style.

All references have been checked and formatted according to Plos one guidelines

18. Figure 1: Flowchart

52+(53+275) =380. You have not accounted for 2 children.

The flow chart has been revised. The 2 children who were initially omitted had died on admission.

19. Figure 2 and Figure 3: Kaplan-Meier plots

The axes should be properly labelled.

The Kaplan Meier plots have been revised.

20. Reviewer #2: The authors use old data (2002-2004) from a small sample (n=330) of hospitalized children under five years old in Malawi. Following these children for 18 months, they compare the death rate in children with severe anaemia to the death rate in children with severe anaemia plus at least moderate acute malnutrition (WHZ < -2). Compared to children with severe anaemia alone, children with severe anaemia plus MAM/SAM were twice as likely to die. The authors conclude by recommending that children who are hospitalized with severe anaemia should also be screened for MAM/SAM.

My primary concern is that the findings are not relevant in today’s context. I may be wrong, but it is hard to believe that hospitalized children are not screened for MAM/SAM. I would guess that weight and height are among the first things to be measured when any child is hospitalized. If a child presents with both MAM/SAM and anaemia, they will receive food and iron supplements, two different treatments. A child who presents with MAM/SAM may also be screened for anaemia. It is hard to believe that MAM/SAM would go completely unnoticed by doctors in today’s context. The authors need to address this and justify the relevance of the study using recent evidence. Other minor comments are below.

Thank you very much for your comment. We acknowledge that the data are from 16 to 18 years ago and the concern that the findings are not relevant in today’s context. However, our study was done for that particular reason. There are no studies, recent or old that have explored survival of children who have both severe anemia and moderate to severe malnutrition despite that these two conditions are often found together. Our study findings are very relevant even in today’s context, because the burden of these two comorbidities remains high and therefore the mortality risk is still very present. We have made revisions in lines 141-142 to make our rationale clearer. However, we agree that the sample size was limited and therefore we have highlighted that this was a limitation in our study and a larger prospective study should be conducted to confirm our findings. This has been highlighted in the discussion section.

21. Abstract

The following sentence is unclear: “Under-five children with severe anaemia were screened and enrolled and of theses children with moderate to severe acute malnutrition; defined as weight-for-height Z-score <-2 were included.” Enrolled and included are distinct? Do you mean included in the analysis? And of course there is a bad typo, “theses”.

This has been revised extensively to be clearer.

22. Don’t use acronym SAM for moderate to severe malnutrition. SAM is severe (HAZ < -3) and does not include moderate. You could say 53 children were identified as having MAM or SAM.

We have revised the manuscript throughout. Moderate to severe malnutrition have been identified as having MAM/SAM.

23. Introduction

The manuscript is missing line numbers.

We have inserted line numbers.

24. The first sentence of the introduction is incorrect. Malnutrition goes much beyond intake. Inadequate or excess intake among other factors can lead to malnutrition.

This is correct. We have revised the definition of malnutrition to “Malnutrition is a complex and multifactorial condition that results from deficiencies, excesses, or imbalances in a person’s intake of energy and/or nutrients and presents in two broad forms; undernutrition which includes wasting, stunting and underweight; and micronutrient deficiencies and obesity.” in lines 81-84.

25. Change ‘commonest’ to most common.

This has been revised in line 99.

26. Methods

Power calculation: can you clarify what the study groups were and what the sample size was for each group? Is it the 53 with MAM/SAM and 275 without MAM/SAM?

We sampled 53 children who had MAM/SAM and 275 children without MAM/SAM. We have revised the section to make it clearer in lines 177-178.

27. What is the acronym AL? Albendazole? 

AL is an acronym for Lumefantrine-Artemether. We have made the correction in line 192-193.

28. Results

Table 1 – groups need to be labeled better. Severe anaemia + MAM/SAM vs. Severe anaemia alone.

Table 1 and 2 has been revised with better labels.

29. Discussion

Do not abbreviate severe anaemia as SA. Avoid abbreviations whenever possible.

We have revised this and spelled severe anemia in full throughout the manuscript.

30. The data are 16-18 years old, not 16 years old.

This is correct. We have made a revision in line 371.

---

## [Decision Letter · Decision Letter 1]

18 Jan 2021

A cohort analysis of survival and outcomes in severely anaemic children with moderate to severe acute malnutrition in Malawi

PONE-D-20-12989R1

Dear Dr. Gondwe,

We’re pleased to inform you that your manuscript has been judged scientifically suitable for publication and will be formally accepted for publication once it meets all outstanding technical requirements.

Kind regards,

Walter R. Taylor

Academic Editor

PLOS ONE

Additional Editor Comments (optional):

Reviewers' comments:

Reviewer's Responses to Questions

**Comments to the Author**

1. If the authors have adequately addressed your comments raised in a previous round of review and you feel that this manuscript is now acceptable for publication, you may indicate that here to bypass the “Comments to the Author” section, enter your conflict of interest statement in the “Confidential to Editor” section, and submit your "Accept" recommendation.

Reviewer #1: All comments have been addressed

2. Is the manuscript technically sound, and do the data support the conclusions?

Reviewer #1: Yes

3. Has the statistical analysis been performed appropriately and rigorously? 

Reviewer #1: Yes

4. Have the authors made all data underlying the findings in their manuscript fully available?

Reviewer #1: Yes

5. Is the manuscript presented in an intelligible fashion and written in standard English?

Reviewer #1: Yes

6. Review Comments to the Author

Reviewer #1: Line# 31; SAM/MAM, most part of the manuscript, MAM/SAM was used. See line#118 and other parts. Keep it consistent.

Line#48; correct to follow-up instead of follow.

Line#81; use lower case for haemoglobin

Line#150; 28.2%, remove space

Line# 155; 10mg/L, use lower case for M.

Line# 168; MAS/SAM?

Lines#189, 191, 193; IRR?

Line#218; “However, this study did not have a comparison group.” It is unclear which study this statement refers to, your study or one of the references?

Line#236, re-admissions

References # 4 and 9 are incomplete.

7. PLOS authors have the option to publish the peer review history of their article (what does this mean?). If published, this will include your full peer review and any attached files.

Reviewer #1: No

---

## [Editor Report · Acceptance letter]

22 Jan 2021

PONE-D-20-12989R1 

A cohort analysis of survival and outcomes in severely anaemic children with moderate to severe acute malnutrition in Malawi 

Dear Dr. Nkosi-Gondwe:

I'm pleased to inform you that your manuscript has been deemed suitable for publication in PLOS ONE. Congratulations! Your manuscript is now with our production department. 

Kind regards, 

on behalf of

Dr. Walter R. Taylor 

Academic Editor

PLOS ONE